# Echocardiographic Probability of Pulmonary Hypertension in Cardiac Surgery Patients—Occurrence and Association with Respiratory Adverse Events—An Observational Prospective Single-Center Study

**DOI:** 10.3390/jcm11195749

**Published:** 2022-09-28

**Authors:** Marta Braksator, Magdalena Jachymek, Karina Witkiewicz, Patrycja Piątek, Wojciech Witkiewicz, Małgorzata Peregud-Pogorzelska, Katarzyna Kotfis, Mirosław Brykczyński

**Affiliations:** 1Department of Cardiology, Pomeranian Medical University, 70-204 Szczecin, Poland; 2Department of Pulmonology, Pomeranian Medical University, 70-204 Szczecin, Poland; 3Department of Anaesthesiology, Intensive Therapy and Acute Intoxications, Pomeranian Medical University, 70-204 Szczecin, Poland; 4Department of Cardiac Surgery, University of Zielona Góra, 65-417 Zielona Góra, Poland

**Keywords:** pulmonary hypertension, LV diastolic dysfunction, postoperative respiratory adverse events, coronary artery bypass grafting, cardiopulmonary bypass, pneumonia

## Abstract

Background: Pulmonary hypertension (PH) is an independent risk factor of increased morbidity and mortality in cardiac surgery patients (CS). The most common cause underlying PH is left ventricular (LV) diastolic dysfunction. This study aimed to evaluate the echocardiographic probability of PH in patients undergoing CS and its correlation with postoperative respiratory adverse events (RAE). Methods: The echocardiographic probability of PH and its correlation with LV diastolic dysfunction was assessed in 56 consecutive adult patients who were qualified for coronary artery bypass grafting (CABG). Later, the postoperative RAE (such as pneumonia, pulmonary congestion, or hypoxemia), the length of intensive care unit (ICU) treatment and mortality in groups with moderate or high (PH-m/h) and low (PH-l) probability of pulmonary hypertension were examined. Results: PH-m/h was observed in 29 patients, of whom 65.5 % had LV diastolic dysfunction stage II or III. A significantly higher occurrence of RAE was observed in the PH-m/h group as compared to the PH-l group. There were no differences between the PH-m/h and PH-l patient groups regarding the in-hospital length of stay or mortality. Conclusions: High or intermediate probability of PH is common in cardiac surgical patients with left ventricular diastolic dysfunction and correlates with respiratory adverse events.

## 1. Introduction

Pulmonary hypertension (PH) is a pathological condition characterized by increased pulmonary vascular pressure. According to the 2015 European Society of Cardiology (ESC)/European Respiratory Society (ERS) guidelines PH is defined as mean pulmonary artery pressure (mPAP) greater than 25 mmHg at rest [1]. The diagnosis of pulmonary hypertension is based on right heart catheterization, but echocardiography is an approved tool to estimate the probability of PH. There are specific algorithms for the detection of PH using echocardiography as a key non-invasive diagnostic test [2,3].

Regardless of its pathophysiology, PH is a significant risk factor for perioperative complications in cardiac and non-cardiac surgery [4,5,6]. Moreover, right ventricular failure, appearing in the natural history of pulmonary hypertension, is a factor that worsens the outcomes [7,8]. In populations undergoing cardiac surgery, when no hemodynamically significant valve abnormalities and no reduced left ventricular ejection fraction are diagnosed, the most common reason for pulmonary hypertension is elevated left ventricular filling pressure due to the left ventricular diastolic dysfunction stage II or III, with a frequency between 27 and 56% [9]. Most studies have examined the influence of elevated left ventricular filling pressures on the length of mechanical ventilation and cardiac adverse events in the population of cardiac surgical patients [9,10]. An examination of changes of the arterial blood gases in patients with PH was provided only in a non-surgical study including patients with pulmonary arterial hypertension (1 type of the disease) [11].

The pathophysiological consequences of elevated pulmonary artery pressure are hypoxemia and hypocapnia [11]. This pathological state may be enhanced by prolonged operation times and the length of extracorporeal circulation in cardiac surgery. Pulmonary congestion, observed commonly in patients with PH due to left heart disease, may promote pulmonary shunt and deepen gasometrical disorders, which can result in a need for prolonged mechanical ventilation and lead to pneumonia. There is a lack of studies concerning the correlation between PH due to LV diastolic dysfunction and respiratory adverse events, such as hypoxemia or pneumonia in CS patients.

The purpose of our study was to investigate whether diastolic heart dysfunction promotes pulmonary hypertension to examine the frequency of high or moderate echocardiographic probability of PH in patients undergoing coronary artery bypass grafting (CABG), and to assess the correlation between PH probability and respiratory adverse events in this population.

## 2. Materials and Methods

This prospective, single-center, observational study was performed between November 2019 and September 2021 in the Department of Cardiac Surgery of the Pomeranian Medical University in Szczecin, Poland. The inclusion criteria are shown in Table 1 and the exclusion criteria are shown in Table 2.

### 2.1. Inclusion and Exclusion Criteria of the Study

Pulmonary hypertension diagnosed according to right heart catheterization (RHC) and treated with targeted therapy, regardless of its type (type 1—pulmonary arterial hypertension, type 4—chronic thromboembolic pulmonary hypertension and type 5—PH with unclear or multifactorial mechanism) was one of the exclusion criteria of the study. The exception was PH type 2 (PH due to left heart disease), but there were no cases of PH type 2 diagnosed preoperatively by RHC in the whole group of patients. We aimed to restrict the hypothetical cause of PH probability observed in our patients to left heart diastolic abnormalities.

### 2.2. Echocardiographic Measurements

All studies were performed by a certified echocardiographer 24 h preoperatively in the Echocardiography Workroom of the Cardiac Surgery Clinic of the Pomeranian Medical University (Philips EpiqCvx; Software Version:3.0.3)—see Appendix A. Postoperative measurements were provided after tracheal extubation within the first three postoperative days.

#### 2.2.1. Probability of Pulmonary Hypertension Assessment

We assessed the probability of pulmonary hypertension in all patients according to the European Society of Cardiology/European Respiratory Society Guidelines for the Diagnosis and Treatment of Pulmonary Hypertension [1]. The evaluation of pulmonary hypertension was based on Tricuspid valve regurgitation maximal Velocity (TR Vmax) and additional echocardiographic PH indicators, which were separated into three categories (A, B, and C):

Category A: right ventricle (RV)/left ventricle (LV) index: RV and LV intracavitary diameter measured in apical four-chamber view, perpendicular to the long axis, at the maximum measurable diameter; LV eccentricity index: ratio of the anterior-inferior and septal-posterolateral cavity dimensions measured at the mid-ventricular level in short-axis parasternal view.

Category B: pulmonic valve acceleration time (PVAccT) lower than 105 m/s or notch; main pulmonary artery diameter and velocity of pulmonary regurgitation in early diastole more than 2.2 m/s.

Category C: inferior vena cava (IVC) diameter; right atrium (RA) area.

Right atrial pressure (RAP) was assessed based on the IVC diameter and its respiratory variation according to the European Society of Echocardiography Guidelines [14]. Right ventricular systolic pressure (sPAP) was defined as the sum of RAP and TR max PG. The scheme of echocardiographic PH assessment is presented in Appendix A—Figure A1. Exemplary image presenting IVC diameter measurement is presented in Appendix A—Figure A2.

#### 2.2.2. Left Ventricular Systolic and Diastolic Function Assessment

The left ventricle ejection fraction was assessed using Simpson’s method. Left ventricular diastolic dysfunction and its stage were defined according to the American Society of Echocardiography and the European Association of Cardiovascular Imaging 2016 Guidelines [15]. Initially, we assessed the left ventricular ejection fraction (LVEF). The diagnosis pathway was dependent on the left ventricular ejection fraction (mildly reduced LVEF: greater than or equal to 40% and less than 50%, or preserved LVEF: greater than or equal to 50%). The scheme of left ventricular diastolic dysfunction assessment is presented in Appendix A—Figure A3.

### 2.3. Perioperative Management and Surgery Procedures

Preoperative management was similar in every patient, with a single dose of oral benzodiazepines one hour before the operation. Drugs from the angiotensin-converting enzymes group and statins were discontinued 24 h before the operation.

#### 2.3.1. General Anesthesia Procedures

In every patient, general anesthesia was performed. After preoxygenation, anesthesia was induced using an intravenous bolus of opioids and etomidate (Etomidate-Lipuro, 0.15–0.3 mg/kg mc). Pancuronium was used as a myorelaxant agent (Pancuronium, Jelfa, 0.1 mg/kg of body weight), inhaled Sevoflurane for maintenance of anesthesia, an intravenous bolus of pancuronium (0.01–0.06 mg/kg of body weight every 30–40 min, according to demand) to maintain the myorelaxant effect. 

#### 2.3.2. Postoperative Management and Measurements

Mechanical ventilation (MV) was provided in the ICU only in pressure-controlled mode with positive end-expiratory pressure (synchronized intermittent mandatory ventilation, with PEEP). The maximal PEEP was 6.0 in every patient during MV. Intravenous infusion of noradrenaline (Levonor, Polfa) was administered to maintain mean arterial pressure greater than 60 mmHg for adequate organ perfusion.

Arterial blood gas measurements were obtained every 240 min after the operation; the first sample was taken directly after ICU admission. The Horowitz Index was measured as a ratio of PaO_2_ in millimeters of mercury and FiO_2_. Pneumonia was defined according to the American Thoracic Society guidelines as the presence of typical changes in chest radiography with at least two of the following: fever, leukocytosis (white blood cell WBC > 12 G/L) or leukopenia (WBC < 4 G/L), or expectoration of pus sputum [16].

ARDS and TRALI were defined according to the American-European Consensus Conference Definition and The National Heart Lung and Blood Institute (NHLBI) Working Group, respectively [17,18]. Empiric antibiotic therapy was administered immediately following the diagnosis of pneumonia. 

### 2.4. Endpoints

We defined the primary endpoints as: pneumonia, pulmonary congestion, the lowest PaO_2_, PaO_2_/FiO_2_, and PaCO_2_ during intubation, the lowest PaO_2_ and PaCO_2_ after extubation and the length of mechanical ventilation.

Secondary endpoints were the length of postoperative ICU stay, length of hospitalization, and occurrence of pneumothorax, pleural effusion, ARDS, TRALI, and in-hospital mortality.

### 2.5. Statistical Analysis

All analyses were performed using Statistica 13 (StatSoft, Inc., Tulsa, OK, USA) software. The continuous variables are presented as mean with standard deviation (SD) or median with interquartile range. The categorical variables are presented as numbers and a percentage. For statistical significance, we used the Student’s *t*-test, U Mann–Whitney test, or Welch test, depending on the distribution or variation. The Chi-square test and Fisher test were used to compare qualitative data. The relationship between the analyzed parameters was evaluated using a multiple linear regression model analysis or logistic regression model using a backward stepwise method. Statistical significance was set at a *p*-value ≤ 0.05.

## 3. Results

### 3.1. Patients’ Characteristic

Sixty-two patients were included in the study. Five patients were excluded postoperatively due to hemorrhagic complications requiring surgical revision and one due to post-CABG myocardial infarction requiring emergency angioplasty. The basic characteristics of both study groups (PH-l and PH-m/h) are presented in Table 3. Patients with a moderate or high probability of pulmonary hypertension were older than patients with a low risk of PH (71 vs. 65 years, *p* = 0.017) and more frequently were women (96.3% of men in PH-l vs. 65.52% of men in PH-m/h, *p* = 0.005), have higher EuroScore II (ESII) (ESII = 0.76 vs. ESII = 1.27, *p* < 0.001). No differences were observed between PH-l and PH-m/h groups for BMI and other comorbidities, including smoking history, COPD frequency, stroke history, arterial hypertension, and diabetes. We did not observe any case of chronic kidney disease stage 3, 4 or 5 (See Table 3).

According to preoperative echocardiography, patients were divided into two categories: low probability of PH (PH-l) and moderate or high probability of PH (PH-m/h). Moderate and high probability of pulmonary hypertension (PH-m/h) was observed in 29 patients (51.7%). Five patients from the PH-m/h group had normal left ventricle diastolic function, and 24 patients had left ventricular diastolic dysfunction, including 19 patients with diastolic dysfunction stages II and III (*p* < 0.001). The comparison of echocardiographic parameters used to assess the risk of pulmonary hypertension is presented in Table 4. As expected, all of them differ significantly between these two groups. We didn’t observe any significant differences between echocardiographic parameters in patients before and after surgical procedure. We didn’t observe any significant differences in heart rate and arterial blood pressure between PH-l and PH-m/h groups. There was no case of hemodynamically significant pericardial effusion—see Table 4.

To find differences between operation types, we compared perfusion, reperfusion, and aortic cross-clamp time, assuming that complex operations are reflected in the prolongation of these times. Perfusion and aortic cross-clamp time were significantly longer in the group with a low probability of PH than in the group with a moderate or high probability of PH (*p* = 0.008; *p* = 0.015, respectively). We did not observe a significant difference in reperfusion time between the PH-l and PH-m/h groups (*p* = 0.09); the results are presented in Table 5.

There was no difference in the rate of postoperative blood transfusion between the PH-l and PH-m/h groups. Norepinephrine was administered in nine patients from the PH-l group and in nine patients from the PH-m/h group (the *p*-value was not significant). We did not observe any need for inotropic support 24 h after the operation. There was no need for the use of Intra-aortic balloon counterpulsation during the perioperative period in the whole group of patients. Postoperative atrial fibrillation was observed in 6 patients with a low probability of PH and 8 patients with a high or intermediate probability of PH.

### 3.2. Primary and Secondary Endpoints

#### 3.2.1. Primary Endpoints

All the primary endpoints and potential complications are shown in Table 6.

Postoperative pneumonia and pulmonary congestion were more frequent in the PH-m/h group than in the PH-l group. The logistic regression model confirmed that moderate and high risk of PH increases the risk of pneumonia and pulmonary congestion in the postoperative period (OR 3.8, 95% CI 1.3–11.1, *p* 0.015, OR 3.29, 95% CI 1.62–6.68, *p* 0.001, respectively). All cases of pneumonia were diagnosed after tracheal extubation and over 48 h after hospital admission. There was no need for reintubation. There were no cases of pulmonary edema.

We did not observe any differences in the minimal and maximal PaCO_2_ (mmHg) between the PH-l and PH-m/h groups during mechanical ventilation nor in the post-extubation period. In the PH-m/h group, we observed a significantly lower minimal and maximal PaO_2_ during mechanical ventilation than in the group with a low probability of PH (*p* = 0.019 and *p* = 0.003, respectively). In the group with moderate or high probability of PH, patients achieved lower maximal PaO_2_ after tracheal extubation than in the low PH risk group (*p* = 0.007), with no significant difference in minimal PaO_2_. The minimal and maximal Horowitz Index was significantly lower in the PH-m/h group than in the PH-l group (*p* = 0.005, *p* = 0.0004, respectively). We did not observe any significant difference in the duration of mechanical ventilation between the two groups (7 h 45 min vs. 8 h 45 min, *p* = 0.1).

Linear regression analysis demonstrated three independent risk factors for hypoxemia during mechanical ventilation: the high or moderate probability of pulmonary hypertension, left ventricular ejection fraction, and BMI (R2 corrected 0.48, *p* < 0.001). Fulfillment of echocardiographic indicators of high or moderate probability of PH decreases PaO_2_ by 171.11 mmHg (Table 7). The effect of PH-m/h probability on minimal PaO_2_ during mechanical ventilation and the paradoxical effect of BMI is shown in Figure 1.

Fulfillment of echocardiographic indicators of high or moderate probability of PH decreases the PaO_2_/FiO_2_ index by 30.71 and BMI over 30 decreases the PaO_2_/FiO_2_ index by 36.81, but adding diastolic dysfunction to the model proves, that heart diastolic dysfunction is a stronger predictor of hypoxemia after cardiac surgery operation, and a moderate or high probability of pulmonary hypertension is no longer statistically significant. The most probable reason is that the PH m/h probability assessed by echocardiography results from diastolic dysfunction, as discussed below (R^2^ corrected 0.33, *p* < 0.001).

#### 3.2.2. Secondary Endpoints

All the secondary endpoints are shown in Table 8.

The length of ICU stay was significantly longer in the moderate to high risk of pulmonary hypertension group than in the low PH risk group (44 vs. 56 h; *p* = 0.016). The linear multivariate regression model was built to determine the influence of PH probability and other co-factors on the length of ICU stay. The only significant factor that influenced the length of the ICU stay was the EuroScore II, but it is responsible for only 23% of the variability of the ICU stay length (R^2^ corrected 0.23, *p* < 0.001).

We did not observe any incidents of acute respiratory distress syndrome (ARDS), TRALI, in-hospital death, pneumothorax, pleural effusion occurrence, and length of in-hospital stay between the PH-l and PH-m/h groups.

## 4. Discussion

In populations undergoing cardiac surgery, when no hemodynamically significant valve abnormalities and no reduced left ventricular ejection fraction exist, the most common reason for pulmonary hypertension is left ventricular elevated filling pressure due to left ventricular diastolic dysfunction stage II or III, with a frequency between 27 and 56% [9,19]. As the disease progresses, elevated filling pressure transfers to the left atrium (LA), leading to its enlargement, then pressure overload, and in the last step, excessive pressure in the pulmonary veins. Over time, chronic lung congestion leads to endothelial proliferation and muscle layer remodeling in pulmonary arteries—this pathological condition leads to pre—postcapillary pulmonary hypertension development, with all clinical symptoms of this disease. With the progression of this pathological state, right ventricle (RV) function decreases [19]. During echocardiography, tricuspid valve regurgitation, RV dilatation and right atrium enlargement are observed [3].

We diagnosed a high or moderate probability of pulmonary hypertension in 29 patients, among whom most had left ventricle diastolic dysfunction stage II or III. These results reflect the pathophysiological mechanism of type 2 PH, which involves the transfer of elevated diastolic pressure from LV to LA and pulmonary veins and facilitates the development of PH [19]. In the study published in 2009, echocardiographic probability of pulmonary hypertension was observed in 83% of patients with heart failure with preserved LV EF (FfpEF) [20]. Furthermore, in a right heart catheterization-based study, pulmonary hypertension was found in 77% of prospectively evaluated patients with HfpEF [7]. We can say that stage of LV diastolic dysfunction correlates with the probability of pulmonary hypertension in patients with mildly reduced and preserved LVEF and without hemodynamically significant valve abnormalities.

The risk factors for PH development in the heart with diastolic dysfunction are age, female sex, and co-morbidity [21]. We assessed preoperative characteristics based on the EuroScore II model, which is a well-recognized preoperative risk score in cardiac surgery [22]. When we examined each risk factor from the EuroScore II separately, we observed significant differences between the PH-l and PH-m/h groups only for age, sex, and systolic pulmonary artery pressure (sPAP) but no differences in co-morbidities, which may result from the strictly defined inclusion and exclusion criteria in our study.

Pneumonia is one of the predominant infectious complications in cardiac surgery and is the most common reason for hospital readmission [23]. A study published in 2021 showed that independent risk factors of postoperative pneumonia in the cardiac surgical population are comorbidities, blood transfusion and long CPB duration, but PH probability was not examined [24].

According to our database, a moderate or high probability of pulmonary hypertension was associated with the high frequency of pneumonia in cardiac surgical patients (11 patients in the PH-m/h group vs. 1 patient in the PH-l group, *p* = 0.002). We conclude that pneumonia is observed at a significant frequency in patients with echocardiographic moderate or high probability of PH, but PH-m/h probability is not likely to be an independent risk factor for pneumonia. All cases of pneumonia were diagnosed after tracheal extubation and over 48 h after hospital admission. According to European and American guidelines suggesting that local expertise should determine the therapy method chosen, in all patients with diagnosed pneumonia, Cephalosporin second generation (BiotraksonPolfa) was administered [16,25]. Sputum probes in patients with pneumonia diagnosed after tracheal extubation were not collected because there is a lack of validation of methods in short-term mechanical ventilation after cardiac surgical procedures. 

The frequency of pulmonary congestion was significantly higher in the group with high or moderate probability of PH than in patients with low PH risk (26 vs. 12 patients, *p* = 0.0008). This observation confirms the pathological mechanism of PH type 2, as described above. 

In the group with a high or moderate probability of PH, we observed significantly lower minimal PaO_2_ during mechanical ventilation than in the group with a low probability of PH (95 vs. 131 mmHg; *p* = 0.019). Moreover, we identified a correlation between the lowest PaO_2_ during mechanical ventilation and three independent factors: body mass index, left ventricular ejection fraction, and a high or moderate probability of PH.

Left ventricular diastolic dysfunction (regardless of its stage) is a stronger predictor of hypoxemia and low PaO_2_/FiO_2_ index after the cardiac operation than high or moderate probability of pulmonary hypertension. The observation confirms the dependence between the probability of PH and diastolic dysfunction. In some cases, echocardiographic indices of PH (for example maximal velocity of tricuspid valve regurgitation) measured in patients with LV diastolic dysfunction were abnormal but didn’t meet the criteria of an echocardiographic high or moderate probability of PH according to the European Society of Cardiology/European Respiratory Society Guidelines for the Diagnosis and Treatment of Pulmonary Hypertension [1]. It may suggest, that in the early stage of LV diastolic dysfunction we observe the right ventricular abnormalities, but we can’t diagnose pulmonary hypertension. However, already in this phase of diastolic dysfunction hypoxemia and pulmonary congestion are observed.

We found an interaction between BMI and moderate or high probability of PH; in patients with a moderate or high probability of PH, BMI over 32 seems to protect from hypoxemia. This phenomenon seems to confirm the obesity paradox, which has been described in numerous studies [26,27].

Hypocapnia is a typical blood gas abnormality for patients with IPAH [11]. We did not observe significant differences between the PH-l and PH-m/h groups in minimal and maximal PaCO_2_. The reason may be that PaCO_2_ depends on minute ventilation, which is adjusted during mechanical ventilation. After extubation, PaCO_2_ was also similar in both groups. However, when it is considered that one of the main reasons for hypocapnia in PAH is low cardiac output, it becomes clear why, in our patients, all of whom had preserved or only mildly reduced LVEF, we did not observe low cardiac output symptoms and found no significant difference in PaCO_2_ after tracheal extubation.

It has been demonstrated that the risk of acute lung injury (ARDS) in the population of cardiac surgical patients is increased. Its occurrence is associated with the use of CPB, massive blood product transfusion, low output cardiac syndrome, and emergency procedures [28,29]. Moreover, transfusion-related acute lung injury (TRALI) is a severe problem after cardiac surgical procedures. We did not observe any cases of ARDS or TRALI, probably due to the lack of massive blood product transfusions, relatively short time of CPB duration in every patient, and elective mode of procedures in our study.

In a retrospective study of patients undergoing mitral- and mitral aortic valve surgery, Pinzani et al. found that preoperative RV failure was a risk factor for perioperative mortality [30]. We did not observe a difference in the average time of mortality between the PH-l and PH-m/h groups. The reason might be that in our patients, despite the presence of PH indicators, we did not observe any cases of significant right ventricular systolic function impairment (TAPSE = 22.89 (±4.21) in the PH-l group and TAPSE = 21.83 (±3.7), *p* = 0.32).

To summarize, a high or intermediate probability of PH is common in patients undergoing cardiac surgery with left ventricular diastolic dysfunction, and it correlates with the occurrence of postoperative respiratory adverse events. To prevent PH development, optimizing pharmacotherapy in patients with LV diastolic dysfunction is crucial. Furthermore, in the future assessment of diastolic dysfunction and elevated LV filling pressure with an evaluation of PH should be considered when developing risk scores in cardiac surgery. Continuous multicenter studies including larger groups of patients, treated according to a standardized protocol are necessary to address the problem further.

### Study Limitations

Our study comes with some limitations. The measurements are based on echocardiography as recommended by the European Society of Cardiology (ESC)/European Respiratory Society (ERS) guidelines and are not confirmed by right heart catheterization. The sample size is relatively small despite the prospective character of the study. In multivariate logistic regression model heart diastolic dysfunction is a stronger predictor of hypoxemia after cardiac surgery operation than the high or moderate probability of pulmonary hypertension. 

The logistic regression model confirmed that moderate and high risk of PH increases the risk of pneumonia and pulmonary congestion in the postoperative period. However, the result should be interpreted with caution because the sample size is too small to calculate multivariate logistic regression.

The COVID-19 pandemic has influenced the number of operations performed in the center. Moreover, this is a single-center study, which may limit the data’s generalizability. The results should be confirmed in another prospective multi-center study to overcome these limitations.

## 5. Conclusions

Our study evaluates the probability of PH, its correlation with heart diastolic function, and its influence on respiratory adverse events in patients with CAD qualified to CABG. We have proven that a high and moderate probability of pulmonary hypertension occurs more often in patients with impaired heart diastolic function, especially with diastolic dysfunction stages II and III. 

We have proven that pneumonia, pulmonary congestion, and hypoxemia in the postoperative period are observed more frequently in patients with a high or moderate probability of pulmonary hypertension, than in patients with a low risk of PH. Moreover, left ventricular diastolic dysfunction is an independent risk factor for lower minimal PaO_2_/FiO_2_ during postoperative mechanical ventilation. It seems that pulmonary hypertension due to left ventricular diastolic dysfunction is a relevant, but still underestimated risk factor in cardiac surgery.

## Figures and Tables

**Figure 1 jcm-11-05749-f001:**
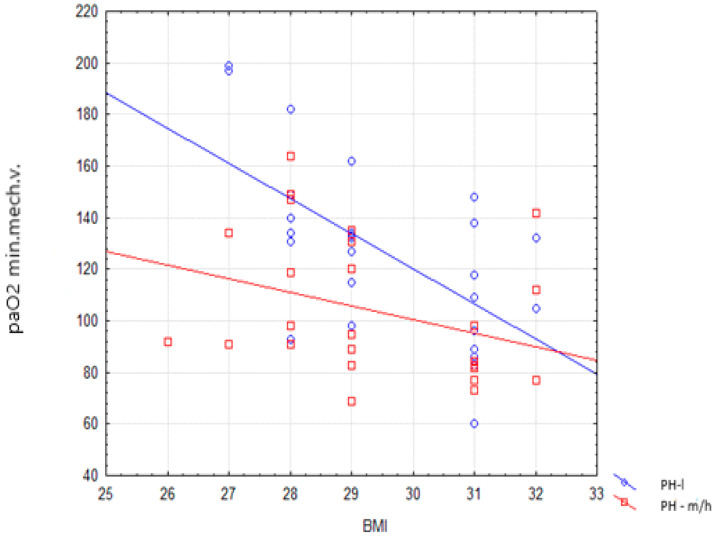
Diagram of dispersion minimal paO_2_ during tracheal intubation and mechanical ventilation (PaO_2_ min. mechanical ventilation) relative to Body Mass Index (BMI) for two categories of echocardiographic PH probability.

**Table 1 jcm-11-05749-t001:** Inclusion criteria of the study.

Inclusion Criteria:
Age > 18 years Coronary artery disease, qualified for coronary artery bypass grafting with use of cardiopulmonary bypass Elective surgery
Left ventricular ejection fraction (LVEF) ≥ 40% in echocardiography

**Table 2 jcm-11-05749-t002:** Exclusion criteria of the study.

Exclusion Criteria:
Pulmonary diseases with severe or moderate restrictive or obstructive disorder
Moderate or severe mitral, tricuspid, aortic or pulmonic valve insufficiency or stenosis, qualified for operation ^1^
Non-elective surgery
Off—pump surgery
LVEF < 40%
Infective endocarditis
Hypertrophic cardiomyopathy
Atrial fibrillation or post pacemaker/cardioverter defibrillator implantation status
Pulmonary arterial hypertension treated with targeted treatment ^2^
Perioperative myocardial infarction (MI type 5) ^3^
Perioperative stroke
Postoperative hemorrhagic complications requiring surgical revision

^1^ According to 2021 ESC/EACTS Guidelines for the management of valvular heart disease [12]. ^2^ Pulmonary hypertension diagnosed preoperatively according to right heart catheterization (regardless of its subtype). ^3^ Perioperative myocardial infarction (MI type 5) was defined according to the fourth universal definition of myocardial infarction [13].

**Table 3 jcm-11-05749-t003:** Preoperative characteristics of patients and type of operation.

	PH-l	PH-m/h	*p*
*n*	27	29	
Age	65 (63–71)	71 (66–73)	0.017
Sex (M, %)	26 (96.3%)	19 (65.52%)	0.005
Obesity	12 (44.44%)	14 (48.28%)	NS
Body mass index	29 (28–31)	29 (28–32)	NS
EuroSCORE II result	0.76 (0.67–0.92)	1.27 (0.93–1.67)	<0.001
Smoking history (*n*, %)	13 (48.15%)	8 (27.59%)	NS
COPD *n* (%)	1 (3.7%)	2 (6.9%)	NS
Stroke *n* (%)	1 (3.7%)	3 (10.34%)	NS
Diabetes *n* (%)	10 (37.04%)	14 (48.28%)	NS
LIMA + RIMA *n* (%)	4 (14.81%)	3 (10.34%)	NS
2 by-pass grafts *n* (%)	12 (44.44%)	15 (51.72%)	NS
3 by-pass grafts *n* (%)	6 (22.22%)	10 (31.48%)	NS
4 by-pass grafts *n* (%)	7 (25.93%)	4 (13.79%)	NS
5 by-pass grafts *n* (%)	2 (7.41%)	0	NS
LAD surgery *n* (%)	27 (100%)	29 (100%)	NS
LCx surgery *n* (%)	12(44,4%)	14 (48,27%)	NS
RCA surgery *n* (%)	19 (70,37%)	20 (68,96%)	NS
Arterial hypertension *n* (%)	21 (77.78%)	22 (75.86%)	NS
CKD stage 2 *n* (%)	9 (33.3%)	10 (34.48%)	NS
CKD stage 3, 4 or 5	0	0	NS

Legend: COPD—chronic obstructive pulmonary disease, LIMA—left internal mammary artery, RIMA—right internal mammary artery, PH-l—low probability of pulmonary hypertension, PH-m/h—moderate or high probability of pulmonary hypertension, CKD—Chronic Kidney Disease LAD—Left anterior descending coronary artery, LCx—Left circumflex coronary artery, RCA—Right coronary artery, NS—not significant.

**Table 4 jcm-11-05749-t004:** Echocardiographic parameters, arterial blood pressure, heart rate, and occurrence of left ventricular diastolic dysfunction before and after CABG.

Parameter	PH-l	PH-m/h	*p*
P Eff pre	0	0	NS
P Eff post	0	0	NS
TR V max pre	1.5 (1.3–1.9)	2.81 (2.2–2.9)	<0.001
TR V max post	1.6 (1.2–1.9)	2.7 (2.0–3.0)	<0.001
TAPSE pre	22.89 (±4.21)	21.83 (±3.7)	NS
TAPSE post	24 (±3.2)	22 (±2.4)	NS
PV AccT pre	108 (92–120)	85 (72–90)	<0.001
PV AccT post	103 (90–115)	80 (71–87)	<0.001
RV/LV pre	0.8 (0.7–0.88)	1.0 (0.9–1.1)	<0.001
RV/LV post	0.8 (0.6–0.9)	1.1 (0.95–1.2)	<0.001
IVC pre	1.2 (1.1–1.7)	2.2 (2.1–2.3)	<0.001
IVC post	1.4 (1.0–1.8)	2.3 (2.0–2.5)	<0.001
sPAP pre	17 (13–21)	47 (31–51)	<0.001
sPAP post	19 (15–22)	48 (30–52)	<0.001
RA area > 18 cm^2^ pre	0 (0%)	10 (34.48%)	<0.001
RA area > 18 cm^2^ post	0 (0%)	11 (37.9%)	<0.001
LVEF pre	63 (55–66)	61.2 (48–67.5)	NS
LVEF post	66 (57–68)	62 (46–69)	NS
LV DD stage II or III pre	5 (18.5%)	19 (65.5%)	<0.001
LV DD stage II or III post	4 (14.8%)	19 (65.5%)	<0.001
SBP pre	147.7 (110–170)	147.4 (105–170)	NS
SBP post	141 (90–165)	139.5 (85–159)	NS
DBP pre	68.8 (60–80)	69.3 (60–85)	NS
DBP post	65 (55–80)	63.4 (52–84)	NS
HR pre	72 (64–90)	76 (68–88)	NS
HR post	79 (72–100)	82 (74–104)	NS

Legend: PH-l—low probability of pulmonary hypertension, PH-m/h—moderate or high probability of pulmonary hypertension, P Eff—Pericardial effusion, TR Vmax—tricuspid valve regurgitation maximal Velocity, TAPSE—Tricuspid Annular Plane Systolic Excursion, PVAccT—pulmonic valve acceleration time, RV/LV—right ventricle/left ventricle diameter index, IVC—inferior vena cava, sPAP—systolic pulmonary artery pressure, RA area—right atrial area, LVEF—left ventricular ejection fraction, SBP—systolic blood pressure, DBP—diastolic blood pressure, HR—heart rate, DD—diastolic dysfunction, pre—preoperative, post—postoperative measurements.

**Table 5 jcm-11-05749-t005:** Cardiopulmonary bypass (CPB) surgery parameters.

CPB Parameters	PH-l	PH-m/h	*p*
Perfusion time (minutes)	59 (52–71)	51 (45–58)	0.008
Reperfusion time (minutes)	21 (16–26)	17 (14–22)	0.09
Aortic cross-clamp time (minutes)	37 (32–45)	32 (27–35)	0.015

Legend: CPB—cardiopulmonary by-pass, PH-l—low probability of pulmonary hypertension, PH-m/h—moderate or high probability of pulmonary hypertension.

**Table 6 jcm-11-05749-t006:** Primary endpoints in PH-m/h and PH-l groups.

Primary Endpoint	PH-l	PH-m/h	*p*
Pneumonia *n* (%)	1 (3.7%)	11 (37.93%)	0.002
Re-intubation	0	0	NS
Pulmonary congestion (%)	12 (44.44%)	26 (89.66%)	0.0008
Pulmonary edema	0	0	NS
PaO_2_ min during mechanical ventilation	131 (98–140)	95.0 (83.0–131.0)	0.019
paO_2_/FiO_2_ min	298 (237–373)	211 (190–291)	0.005
PaCO_2_ min during mechanical ventilation	40 (37–44)	40.5 (35.5–43.5)	NS
PaO_2_ min after tracheal extubation	110 (93–135)	100 (85–113)	NS
PaCO_2_ min after tracheal extubation	42.0 (39–44)	43 (39.5–44.5)	NS
Length of mechanical ventilation (hours)	7.45 (5.00–9.05)	8.45 (6.15–10.40)	NS

Legend: PH-l—low probability of pulmonary hypertension, PH-m/h—moderate or high probability of pulmonary hypertension, PaO_2_ min = the lowest PaO_2_, PaO_2_/FiO_2_ min = the lowest Horowitz index during mechanical ventilation; PaCO_2_ min = the lowest PaCO_2_, NS—not significant.

**Table 7 jcm-11-05749-t007:** Multivariate linear regression for PaO_2_ during operation.

Variables	Co-Efficient (®)	95% CI	*p*
Intercept	229.85	84.8–374.88	0.002
PH m-h	−171.11	−291.08–−51.15	0.006
LV EF	1.38	0.68–2.08	<0.001
BMI	−6.67	−10.96–−2.41	0.003
PH and BMI interaction	5.43	1.37–9.49	0.009

Legend: CI—Confidence interval, PH-m/h—moderate or high probability of pulmonary hypertension, LVEF—left ventricular ejection fraction, BMI—body mass index, PH—pulmonary hypertension.

**Table 8 jcm-11-05749-t008:** Secondary endpoints in PH-m/h and PH-l groups.

Secondary Endpoint	PH-l	PH-m/h	*p*
Pneumothorax	1 (3.7%)	1 (3.45%)	NS
Pleural effusion	1 (3.7%)	2 (6.9%)	NS
ARDS	0	0	NS
TRALI	0	0	NS
Length of ICU stay (hours)	44 (36–54)	56 (40–72)	0.016
Length of hospitalization (days)	7 (6–9)	7 (6–8)	NS
In-hospital mortality	0	0	NS

Legend: PH-l—low probability of pulmonary hypertension, PH-m/h—moderate or high probability of pulmonary hypertension, NS—not significant, ARDS—Acute Respiratory Distress Syndrome, TRALI—Transfusion—Related Acute Lung Injury, ICU—Intensive Care Unit.

## Data Availability

The raw data will be available from the corresponding author upon a reasonable request.

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
