# Peer review of "Echocardiographic Probability of Pulmonary Hypertension in Cardiac Surgery Patients—Occurrence and Association with Respiratory Adverse Events—An Observational Prospective Single-Center Study"

_jcm, 2022, doi:10.3390/jcm11195749_

Round 1

Reviewer 1 Report (New Reviewer)

Dear authors,

Below, you will find my comments related to the article: 

- What were the parameters used to define moderate or high (PH-m/h) and low (PH–l) probability of pulmonary hypertension?

- How did the authors define that pulmonary hypertension was caused by left ventricular dysfunction rather than a post-CABG condition? Same for the outcomes: could we say that the outcomes were more related to PH or to a post-CABG state? 

- How long before/after CABG were the echo parameters evaluated?

- Why were only CABG patients included?

- Why was the sample size calculated to have sufficient power to assess the primary and secondary endpoints?

Author Response

Dear Reviewer,

Thank you very much for the revision. We appreciate all the comments regarding our manuscript, as they make it stronger. We responded in a detailed point-to-point manner.

Q: What were the parameters used to define moderate or high (PH-m/h) and low (PH–l) probability of pulmonary hypertension?

A: Thank you for your question. We assessed the probability of pulmonary hypertension in all patients according to the European Society of Cardiology/European Respiratory Society Guidelines for the Diagnosis and Treatment of Pulmonary Hypertension 2015. First, we evaluated Tricuspid valve regurgitation maximal velocity (TR Vmax) and additional echocardiographic PH indicators. When TR Vmax was assessed as less than or equal to 2,8 m/s or impossible to measure and no additional PH indicators (such as right atrium or inferior vena cava enlargement  - as described in the manuscript) echocardiographic probability of pulmonary hypertension was low.  When TR Vmax was greater than 3,4 m/s, or TR Vmax was 2,9 – 3,4 m/s with additional PH indicators, echocardiographic probability of PH was high. When TR Vmax was greater than or equal to 2,8 m/s and additional PH indicators were observed, or TR Vmax was 2,9 – 3,4 m/s without additional PH indicators, the risk or PH was moderate. 

We added the scheme of echocardiographic PH assessment and exemplary image from our data to Appendix A. 

Q: How did the authors define that pulmonary hypertension was caused by left ventricular dysfunction rather than a post-CABG condition? Same for the outcomes: could we say that the outcomes were more related to PH or to a post-CABG state?

A: Thank you for your valuable comment. The first assessment of PH and heart diastolic function was provided before the operation. During post-CABG period we provided additional echocardiographic measurements after tracheal extubation to eliminate possible  confounding factors, i.e. parameters related to mechanical ventilation, such as positive–end expiratory pressure (PEEP). We did not find any significant differences in the results of echocardiographic measurements before and after CABG in both groups – we added this data to the manuscript and modified Table 4. 

Q: How long before/after CABG were the echo parameters evaluated?

A: Thank you for your question. All echo parameters were evaluated 24 hours before and by the third postoperative day after CABG (time of mechanical ventilation was excluded ). The long–term follow–up was impossible to organizational issues associated with the  SARS-CoV-2 pandemic.

Q: Why were only CABG patients included?

A: Thank you for this question. We wanted to limit the hypothetical cause of pulmonary hypertension probability observed in our patients due to left heart diastolic abnormalities (one of our aims was to assess the correlation between left ventricular diastolic dysfunction and the probability of pulmonary hypertension). It was assumed that hemodynamically significant valve abnormalities might be the confounding factors.

Q: Why was the sample size calculated to have sufficient power to assess the primary and secondary endpoints?

A: Thank you for your question. We agree, that the statistical analysis gains on reliability when larger number of patients are enrolled into the study. It is most visible when we compare complication rates, because there are too few observations to determine the real effect of increased PH risk on the probability of postoperative complications. Nevertheless, we could not calculate the number of patients needed for the study basing the assumption on continuous endpoints (such as paO2 for instance), as there were no previous studies in this topic. Yet, we have reached statistical significancy in this part. The study was limited in time and was proceeded during the SARS-CoV-2 pandemic when there were fewer patients for planned operations. The further analysis regarding the influence of PH probability on postoperative complications is necessary.

With best regards

Reviewer 2 Report (New Reviewer)

This study showed that pulmonary hypertension(PH) in echocardiography might be a risk factor of cardiac surgery. The results suggest that the patients with moderate or high probability of PH had more events associated with respiratory compared with the patients with low probability of PH. Overall the results are interesting, but the data was descriptive.

Major concerns

1)    Were their surgeries same methods? Please show the ratio of on pump CABG and off pump CABG. Moreover, please show the ratio of coronary artery, such as LAD, LCX or RCA.

2) The left ventricle diastolic dysfunction might be changed after CABG and it is important for the occurence of adverse event after surgery. Thus, please show the echocardiographic-parameter of left ventricle diastolic dysfunction and estimated PAP after surgery.

3) PH-m/h groups had more occurrence of pneumonia or pulmonary congestion compared with PH-l groups. On the other hand, the duration of mechanical ventilation was comparable between these groups. There are some gaps. Was the pneumonia or pulmonary congestion mild?

4) Why was the duration of ICU longer in PH-m/h groups than in PH-l groups? The difference was very short time such as 12 hours.

Author Response

Dear Reviewer,

Thank you very much for the revision. We appreciate all the comments regarding our manuscript, as they make it stronger. We responded in a detailed point-to-point manner.

This study showed that pulmonary hypertension (PH) in echocardiography might be a risk factor of cardiac surgery. The results suggest that the patients with moderate or high probability of PH had more events associated with respiratory compared with the patients with low probability of PH. Overall the results are interesting, but the data was descriptive.

1)    Were their surgeries same methods? Please show the ratio of on pump CABG and off pump CABG. Moreover, please show the ratio of coronary artery, such as LAD, LCX or RCA.

Answer: Thank you for your valuable comment. Off – pump surgery was one of the exclusion criteria of the study. We added the data to Table 3 in our manuscript. According to your suggestion, we modified Table 3, adding the information concerning individual coronary arteries surgery.

2) The left ventricle diastolic dysfunction might be changed after CABG, and it is important for the occurrence of adverse event after surgery. Thus, please show the echocardiographic parameter of left ventricle diastolic dysfunction and estimated PAP after surgery.

Answer: Thank you for your valuable comment. We added the information about pre – and postoperative occurrence of left ventricular diastolic dysfunction to the Table 4 in our manuscript. Also, we added the scheme of LV diastolic dysfunction assessment to Appendix A. 

3) PH-m/h groups had more occurrence of pneumonia or pulmonary congestion compared with PH-l groups. On the other hand, the duration of mechanical ventilation was comparable between these groups. There are some gaps. Was the pneumonia or pulmonary congestion mild?

Answer: Thank you for your question. All cases of pneumonia cases in our patients were diagnosed after tracheal extubation and over 48 hours after hospital admission. We didn’t observe any cases of acute respiratory failure. There was no need to re-intubation. Pneumonia was defined according to the American Thoracic Society guidelines as the presence of typical changes in chest radiography with at least two of the following: fever, leukocytosis (white blood cell WBC>12 G/l) or leukopenia (WBC<4 G/l), or expectoration of pus sputum. Pulmonary congestion was recognized according to chest radiogram. We didn’t observe any case of pulmonary edema. We added this information to our manuscript.

4) Why was the duration of ICU longer in PH-m/h groups than in PH-l groups? The difference was very short time such as 12 hours.

Answer: Thank you for your question. Duration of ICU stay was longer in PH – m/h group due to hypoxemia. Some patients required extended oxygen therapy and continuous monitoring of vital signs. Inotropic support 24 hours after the operation was not required in any patients. Surgical complications requiring revision were part of the exclusion criteria of this study. 

With best regards

Round 2

Reviewer 1 Report (New Reviewer)

Thanks for your clarifications.

Author Response

Thank you for accepting our clarifications.

Reviewer 2 Report (New Reviewer)

All of my concerns are addressed.

Author Response

Thank you for your kind assessment of our manuscript.

This manuscript is a resubmission of an earlier submission. The following is a list of the peer review reports and author responses from that submission.

Round 1

Reviewer 1 Report

Braksator et al conducted a prospective cohort study investigating the associations between the echocardiographic probability of pulmonary hypertension (PH), left ventricular (LV) diastolic function, and the development of postoperative respiratory adverse events. The authors observed that a high or intermediate probability of PH was associated with LV diastolic dysfunction, as well as respiratory adverse events. However, this study was subject to major limitations listed below.

  1. Although this study was claimed to be a prospective cohort study, the totally enrolled patient number was only 56. The study cohort could be too small to reach a robust statistical significance. Since adult patients with coronary artery diseases requiring coronary artery bypass grafting surgery are not rare clinically, more patients should be prospectively enrolled.

  2. In statistical analysis, the authors wrote “the relationship between the analyzed parameters was evaluated using multiple regression model analysis.” However, what kind of regression analysis was used was not clearly described, ex. Linear, logistic, or Cox regression analysis?

  3. The key analysis in the present study should be the results of logistic regression analysis. However, there was a lack of a clear table demonstrating the results of logistic regression analysis.

  4. The writing of the section of Results should be more organized. One single sentence should not be presented as one paragraph, which would make the information fragmented and harder to read.

  5. The association between LV diastolic dysfunction and the presence of PH is not a novel finding.

  6. The descriptions in the Results seemed to have some major errors. For instance: 

  1. P.8, line 203: The comparison of pre-operative baseline characteristics was in Table 7, rather than Table 2.

  2. P.8, line 205: It seemed that patients in the group of PH-l were more likely to be men (96.3% as shown in Table 7), but rather women.

  1. There was a lack of “study limitations” in this manuscript, which could be an essential part that should be included at the end of the Discussion.

  2. English grammar needs extensive improvement throughout the article.

Author Response

Dear Reviewer,

We would like to thank you for a thorough examination of our manuscript and the valuable comments. We have corrected our manuscript and presented the answers below:

  1. Although this study was claimed to be a prospective cohort study, the totally enrolled patient number was only 56. The study cohort could be too small to reach a robust statistical significance. Since adult patients with coronary artery diseases requiring coronary artery bypass grafting surgery are not rare clinically, more patients should be prospectively enrolled.

Answer: We agree that the statistical analysis gains on reliability when larger number of patients are enrolled into the study. It is most visible when we compare complication rates because there are too few observations to determine the real effect of increased PH risk on the probability of postoperative complications. We couldn’t calculate the number of patients needed for the study basing on continuous endpoints (such as paO2 for instance), as there were no previous studies in this topic. Yet, we have reached significancy in this part.

The study was limited in time and proceeded during the COVID-19 pandemic, when there were fewer patients for planned operations. Thus, we agree that further analysis regarding the influence of PH probability on postoperative complications is necessary.

  1. In statistical analysis, the authors wrote “the relationship between the analyzed parameters was evaluated using multiple regression model analysis.” However, what kind of regression analysis was used was not clearly described, ex. Linear, logistic, or Cox regression analysis?

Answer: Thank you for this remark. The information in the manuscript was not complete. It was linear regression model. We already made suggested change in the manuscript. We also added the information about logistic regression, which also was not present in statistical analysis section.

  1. The key analysis in the present study should be the results of logistic regression analysis. However, there was a lack of a clear table demonstrating the results of logistic regression analysis.

Answer: Thank you for this suggestion. We used the logistic regression only ones, when we determine if PH probability is an independent factor of pneumonia in postoperative period. Note, that we have changed the result, because we calculated it again with a different method. To perform the multivariate logistic regression, we would need a bigger group (at least 100 patients for 2 variables).

Other primary endpoint, that is lowest paO2, paO2/FiO2 and paCO2 and their relationship to PH probability and other cofactors are calculated as linear regression model, as the dependent variable (paO2) is continuous. We also created linear multiple regression model for pO2/FiO2 as a dependent variable, but the results were analogous to pO2 (with the significant influence of pH probability, BMI, and LV EF on paO2/FiO2). The Horowitz index in derived from paO2, so we decided to not include this result, as in do not bring any new information.

To increase the clarity, we put regression results in a table.

  1. The writing of the section of Results should be more organized. One single sentence should not be presented as one paragraph, which would make the information fragmented and harder to read.

Answer: Thank you for your suggestion. We have reorganized that section.

  1. The association between LV diastolic dysfunction and the presence of PH is not a novel finding.

Answer: We are aware of that. It wasn’t the purpose of this study to find that correlation. The aim of our study was to assess the occurrence of echocardiographic high and moderate probability of PH in cardiac surgery patients and to investigate the association between probability of PH and postoperative respiratory adverse events, especially gasometrical abnormalities, and pneumonia. 

  1. The descriptions in the Results seemed to have some major errors. For instance: 

P.8, line 203: The comparison of pre-operative baseline characteristics was in Table 7, rather than Table 2. -

Answer: Thank you for pointing it out – we have corrected this mistake.

P.8, line 205: It seemed that patients in the group of PH-l were more likely to be men (96.3% as shown in Table 7), but rather women.

Answer: Thank you for pointing it out – we have corrected this mistake.

  1. There was a lack of “study limitations” in this manuscript, which could be an essential part that should be included at the end of the Discussion.

Answer: Thank you for pointing it out – we have corrected this mistake.

  1. English grammar needs extensive improvement throughout the article.

Answer: Thank you for your suggestion. We have edited the whole text of the manuscript.

With best regards

Reviewer 2 Report

the paper is intriguing and well written. There is a lack of procedural data which might potentially impact on study outcomes; namely there is no mention on: number and types of grafts, pattern of release post procedural specific myocardial injury markers, usage of IABP or need for prolonged inotropic support, rates of stroke, postoperative AF. All in all, Authors might add these data and comment of them.

Author Response

Dear Reviewer,

We would like to thank you for a thorough examination of our manuscript and the valuable comments. We have corrected our manuscript and presented the answers below:

The paper is intriguing and well written. There is a lack of procedural data which might potentially impact on study outcomes; namely there is no mention on: number and types of grafts, pattern of release post procedural specific myocardial injury markers, usage of IABP or need for prolonged inotropic support, rates of stroke, postoperative AF. All in all, Authors might add these data and comment of them.

Answers:

Perioperative myocardial infarction (MI type 5) was defined according to fourth universal definition of myocardial infarction. When diagnosed, myocardial infarction type 5 was one of exclusion criteria of the study.

Perioperative stroke was one of exclusion criteria of the study.

Prolonged inotropic support was defined as the need for inotropic support 24 hours after operation. We didn’t observe any case of need for inotropic support more than 24 hours after the operation.

There was no need for use of Intra-aortic balloon counterpulsation in perioperative period in the whole group of patients.

Postoperative atrial fibrillation was observed in 6 patients with low probability of PH and in 8 patients with high or intermediate probability of PH. We added this information on table 8.

We added the information about number of by-pass grafts and statistical analysis results on table 6.

With best regards

Round 2

Reviewer 1 Report

The authors didn't improve the article too much. Additionally, several English errors could still be found throughout the whole manuscript. 

In general, I would suggest the authors shorten the manuscript from method to discussion. 

For example, Table 3, Figure 1, Figure 2, Figure 3, and Figure 4 are not necessary since these are the basic concept of echocardiography. The authors only need to cite the references. 

Table 4 could be written in the description. 

Try to merge Tables 5-8 into 1-2 tables and reserve the relatively important information. 

Discussion is too wordy. The authors need to provide more clinical implications of the present findings. 

Author Response

Dear Reviewer,

Thank you very much for the second round of revision. We appreciate all the comments as they make our manuscript stronger. We responded in a detailed point-to-point manner.

  1. The authors didn't improve the article too much. Additionally, several English errors could still be found throughout the whole manuscript. 

Answer: Thank you for this comment and an option to improve the paper. We corrected English errors throughout the whole manuscript. 

  1. In general, I would suggest the authors shorten the manuscript from method to discussion. 

Answer This has been done, as suggested – please see below.

  1. For example, Table 3, Figure 1, Figure 2, Figure 3, and Figure 4 are not necessary since these are the basic concept of echocardiography. The authors only need to cite the references. 

Answer: We removed Table 3, Figure 1, Figure 2, Figure 3, and Figure 4 and cited the references. 

  1. Table 4 could be written in the description. 

Answer: We removed Table 4 as it was already described in the text.

  1. Try to merge Tables 5-8 into 1-2 tables and reserve the relatively important information. 

Answer: Thank you for this comment. It was not possible to merge all the information, as the other reviewer requested additional data, but following your advice we removed table 8 and described the findings in the text.

  1. Discussion is too wordy. The authors need to provide more clinical implications of the present findings. 

Answer: Thank you for this suggestion, we tried to shorten the length of the discussion and make it more focused.

With best regards

Reviewer 2 Report

Significantly improved, suitable for publication 

Author Response

Thank you for this kind comment. It is very much appreciated.